# MDP Homomorphic Networks:
# Group Symmetries in Reinforcement Learning

**Elise van der Pol**
UvA-Bosch Deltalab
University of Amsterdam
e.e.vanderpol@uva.nl

**Daniel E. Worrall**
Philips Lab
University of Amsterdam
d.e.worrall@uva.nl

**Herke van Hoof**
UvA-Bosch Deltalab
University of Amsterdam
h.c.vanhoof@uva.nl

**Frans A. Oliehoek**
Department of Intelligent Systems
Delft University of Technology
f.a.oliehoek@tudelft.nl

**Max Welling**
UvA-Bosch Deltalab
University of Amsterdam
m.welling@uva.nl

## Abstract

This paper introduces MDP homomorphic networks for deep reinforcement learning. MDP homomorphic networks are neural networks that are equivariant under *symmetries* in the joint state-action space of an MDP. Current approaches to deep reinforcement learning do not usually exploit knowledge about such structure. By building this prior knowledge into policy and value networks using an equivariance constraint, we can reduce the size of the solution space. We specifically focus on group-structured symmetries (invertible transformations). Additionally, we introduce an easy method for constructing equivariant network layers numerically, so the system designer need not solve the constraints by hand, as is typically done. We construct MDP homomorphic MLPs and CNNs that are equivariant under either a group of reflections or rotations. We show that such networks converge faster than unstructured baselines on CartPole, a grid world and Pong.

## 1 Introduction

This paper considers learning decision-making systems that exploit symmetries in the structure of the world. Deep reinforcement learning (DRL) is concerned with learning neural function approximators for decision making strategies. While DRL algorithms have been shown to solve complex, high-dimensional problems [35, 34, 26, 25], they are often used in problems with large state-action spaces, and thus require many samples before convergence. Many tasks exhibit symmetries, easily recognized by a designer of a reinforcement learning system. Consider the classic control task of balancing a pole on a cart. Balancing a pole that falls to the right requires an *equivalent*, but mirrored, strategy to one that falls to the left. See Figure 1. In this paper, we exploit knowledge of such symmetries in the state-action space of Markov decision processes (MDPs) to reduce the size of the solution space.

We use the notion of *MDP homomorphisms* [32, 30] to formalize these symmetries. Intuitively, an MDP homomorphism is a map between MDPs, preserving the essential structure of the original MDP, while removing redundancies in the problem description, i.e., equivalent state-action pairs. The removal of these redundancies results in a smaller state-action space, upon which we may more easily build a policy. While earlier work has been concerned with discovering an MDP homomorphism for a given MDP [32, 30, 27, 31, 6, 39], we are instead concerned with how to construct deep policies, satisfying the MDP homomorphism. We call these models *MDP homomorphic networks*.

MDP homomorphic networks use experience from one state-action pair to improve the policy for all 'equivalent' pairs. See Section 2.1 for a definition. They do this by tying the weights for two states if they are equivalent under a transformation chosen by the designer, such as $s$ and $L[s]$ in Figure 1. Such weight-tying follows a similar principle to the use of convolutional networks [18], which are equivariant to translations of the input [11]. In particular, when equivalent state-action pairs can be related by an invertible transformation, which we refer to as *group-structured*, we show that the policy network belongs to the class of *group-equivariant neural networks* [11, 46].Equivariant neural networks are a class of neural network, which have built-in symmetries [11, 12, 46, 43, 41]. They are a generalization of convolutional neural networks—which exhibit translation symmetry—to transformation groups (group-structured equivariance) and transformation semigroups [47] (semigroup-structured equivariance). They have been shown to reduce sample complexity for classification tasks [46, 44] and also to be universal approximators of symmetric functions[1] [48]. We borrow from the literature on group equivariant networks to design policies that tie weights for state-action pairs given their equivalence classes, with the goal of reducing the number of samples needed to find good policies. Furthermore, we

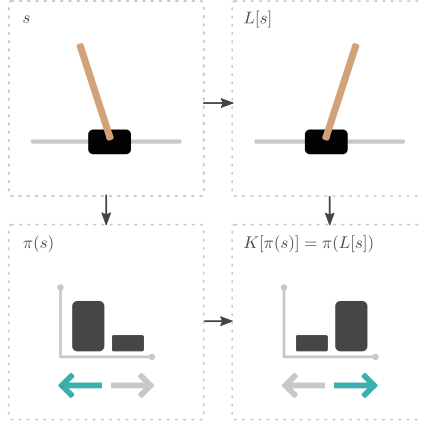

Figure 1: Example state-action space symmetry. Pairs $(s, \leftarrow)$ and $(L[s], \rightarrow)$ (and by extension $(s, \rightarrow)$ and $(L[s], \leftarrow)$) are symmetric under a horizontal flip. Constraining the set of policies to those where $\pi(s, \leftarrow) = \pi(L[s], \rightarrow)$ reduces the size of the solution space.

can use the MDP homomorphism property to design not just policy networks, but also value networks and even environment models. MDP homomorphic networks are agnostic to the type of model-free DRL algorithm, as long as an appropriate transformation on the output is given. In this paper we focus on equivariant policy and invariant value networks. See Figure 1 for an example policy.

An additional contribution of this paper is a novel numerical way of finding equivariant layers for arbitrary transformation groups. The design of equivariant networks imposes a system of linear constraint equations on the linear/convolutional layers [12, 11, 46, 43]. Solving these equations has typically been done analytically by hand, which is a time-consuming and intricate process, barring rapid prototyping. Rather than requiring analytical derivation, our method only requires that the system designer specify input and output transformation groups of the form {state transformation, policy transformation}. We provide Pytorch [29] implementations of our equivariant network layers, and implementations of the transformations used in this paper. We also experimentally demonstrate that exploiting equivalences in MDPs leads to faster learning of policies for DRL.

Our contributions are two-fold:

- We draw a connection between MDP homomorphisms and group equivariant networks, proposing MDP homomorphic networks to exploit symmetries in decision-making problems;
- We introduce a numerical algorithm for the automated construction of equivariant layers.

## 2 Background

Here we outline the basics of the theory behind MDP homomorphisms and equivariance. We begin with a brief outline of the concepts of equivalence, invariance, and equivariance, followed by a review of the Markov decision process (MDP). We then review the MDP homomorphism, which builds a map between 'equivalent' MDPs.

### 2.1 Equivalence, Invariance, and Equivariance

**Equivalence** If a function $f : \mathcal{X} \rightarrow \mathcal{Y}$ maps two inputs $x, x' \in \mathcal{X}$ to the same value, that is $f(x) = f(x')$, then we say that $x$ and $x'$ are $f$-equivalent. For instance, two states $s, s'$ leading to the

same optimal value $V^*(s) = V^*(s')$ would be $V^*$-equivalent or *optimal value equivalent* [30]. An example of two optimal value equivalent states would be states $s$ and $L[s]$ in the CartPole example of Figure 1. The set of all points $f$-equivalent to $x$ is called the *equivalence class* of $x$.

**Invariance and Symmetries** Typically there exist very intuitive relationships between the points in an equivalence class. In the CartPole example of Figure 1 this relationship is a horizontal flip about the vertical axis. This is formalized with the transformation operator $L_g : \mathcal{X} \to \mathcal{X}$, where $g \in G$ and $G$ is a mathematical group. If $L_g$ satisfies

$$f(x) = f(L_g[x]), \qquad \text{for all } g \in G, x \in \mathcal{X}, \tag{1}$$

then we say that $f$ is *invariant* or *symmetric* to $L_g$ and that $\{L_g\}_{g \in G}$ is a set of *symmetries* of $f$. We can see that for the invariance equation to be satisfied, it must be that $L_g$ can only map $x$ to points in its equivalence class. Note that in abstract algebra for $L_g$ to be a true transformation operator, $G$ must contain an identity operation; that is $L_g[x] = x$ for some $g$ and all $x$. An interesting property of transformation operators which leave $f$ invariant, is that they can be composed and still leave $f$ invariant, so $L_g \circ L_h$ is also a symmetry of $f$ for all $g, h \in G$. In abstract algebra, this property is known as a *semigroup property*. If $L_g$ is always invertible, this is called a *group property*. In this work, we experiment with group-structured transformation operators. For more information, see [14]. One extra helpful concept is that of *orbits*. If $f$ is invariant to $L_g$, then it is invariant along the orbits of $G$. The orbit $\mathcal{O}_x$ of point $x$ is the set of points reachable from $x$ via transformation operator $L_g$:

$$\mathcal{O}_x \triangleq \{L_g[x] \in \mathcal{X} | g \in G\}. \tag{2}$$

**Equivariance** A related notion to invariance is *equivariance*. Given a transformation operator $L_g : \mathcal{X} \to \mathcal{X}$ and a mapping $f : \mathcal{X} \to \mathcal{Y}$, we say that $f$ is equivariant [11, 46] to the transformation if there exists a second transformation operator $K_g : \mathcal{Y} \to \mathcal{Y}$ in the output space of $f$ such that

$$K_g[f(x)] = f(L_g[x]), \qquad \text{for all } g \in G, x \in \mathcal{X}. \tag{3}$$

The operators $L_g$ and $K_g$ can be seen to describe the same transformation, but in different spaces. In fact, an equivariant map can be seen to map orbits to orbits. We also see that invariance is a special case of equivariance, if we set $K_g$ to the identity operator for all $g$. Given $L_g$ and $K_g$, we can solve for the collection of equivariant functions $f$ satisfying the equivariance constraint. Moreover, for linear transformation operators and linear $f$ a rich theory already exists in which $f$ is referred to as an *intertwiner* [12]. In the equivariant deep learning literature, neural networks are built from interleaving intertwiners and equivariant nonlinearities. As far as we are aware, most of these methods are hand-designed per pair of transformation operators, with the exception of [13]. In this paper, we introduce a computational method to solve for intertwiners given a pair of transformation operators.

## 2.2 Markov Decision Processes

A Markov decision process (MDP) is a tuple $(\mathcal{S}, \mathcal{A}, R, T, \gamma)$, with *state space* $\mathcal{S}$, *action space* $\mathcal{A}$, *immediate reward function* $R : \mathcal{S} \times \mathcal{A} \to \mathbb{R}$, *transition function* $T : \mathcal{S} \times \mathcal{A} \times \mathcal{S} \to \mathbb{R}_{\geq 0}$, and *discount factor* $\gamma \in [0, 1]$. The goal of solving an MDP is to find a policy $\pi \in \Pi$, $\pi : \mathcal{S} \times \mathcal{A} \to \mathbb{R}_{\geq 0}$ (written $\pi(a|s)$), where $\pi$ normalizes to unity over the action space, that maximizes the expected return $R_t = \mathbb{E}_\pi[\sum_{k=0}^{T} \gamma^k r_{t+k+1}]$. The expected return from a state $s$ under a policy $\pi$ is given by the *value function* $V^\pi$. A related object is the $Q$-value $Q^\pi$, the expected return from a state $s$ after taking action $a$ under $\pi$. $V^\pi$ and $Q^\pi$ are governed by the well-known Bellman equations [5] (see Supplementary). In an MDP, optimal policies $\pi^*$ attain an optimal value $V^*$ and corresponding $Q$-value given by $V^*(s) = \max_{\pi \in \Pi} V^\pi(s)$ and $Q^*(s) = \max_{\pi \in \Pi} Q^\pi(s)$.

**MDP with Symmetries** Symmetries can appear in MDPs. For instance, in Figure 2 CartPole has a reflection symmetry about the vertical axis. Here we define an *MDP with symmetries*. In an MDP with symmetries there is a set of transformations on the state-action space, which leaves the reward function and transition operator invariant. We define a state transformation and a state-dependent action transformation as $L_g : \mathcal{S} \to \mathcal{S}$ and $K_g^s : \mathcal{A} \to \mathcal{A}$ respectively. Invariance of the reward function and transition function is then characterized as

$$R(s, a) = R(L_g[s], K_g^s[a]) \qquad \text{for all } g \in G, s \in \mathcal{S}, a \in \mathcal{A} \tag{4}$$

$$T(s'|s, a) = T(L_g[s']|L_g[s], K_g^s[a]) \qquad \text{for all } g \in G, s \in \mathcal{S}, a \in \mathcal{A}. \tag{5}$$

Written like this, we see that in an MDP with symmetries the reward function and transition operator are invariant along orbits defined by the transformations $(L_g, K_g^s)$.

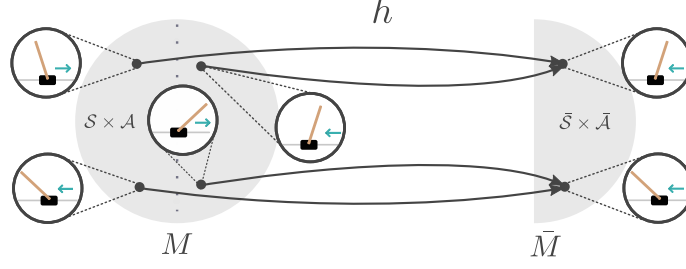

Figure 2: Example of a reduction in an MDP's state-action space under an MDP homomorphism $h$. Here 'equivalence' is represented by a reflection of the dynamics in the vertical axis. This equivalence class is encoded by $h$ by mapping all equivalent state-action pairs to the same abstract state-actions.

**MDP Homomorphisms**   MDPs with symmetries are closely related to MDP homomorphisms, as we explain below. First we define the latter. An *MDP homomorphism* $h$ [32, 30] is a mapping from one MDP $M = (\mathcal{S}, \mathcal{A}, R, T, \gamma)$ to another $\bar{M} = (\bar{\mathcal{S}}, \bar{\mathcal{A}}, \bar{R}, \bar{T}, \gamma)$ defined by a surjective map from the state-action space $\mathcal{S} \times \mathcal{A}$ to an *abstract state-action space* $\bar{\mathcal{S}} \times \bar{\mathcal{A}}$. In particular, $h$ consists of a tuple of surjective maps $(\sigma, \{\alpha_s | s \in \mathcal{S}\})$, where we have the state map $\sigma : \mathcal{S} \to \bar{\mathcal{S}}$ and the state-dependent action map $\alpha_s : \mathcal{A} \to \bar{\mathcal{A}}$. These maps are built to satisfy the following conditions

$$\bar{R}(\sigma(s), \alpha_s(a)) \triangleq R(s, a) \qquad\qquad \text{for all } s \in \mathcal{S}, a \in \mathcal{A}, \qquad (6)$$

$$\bar{T}(\sigma(s') | \sigma(s), \alpha_s(a)) \triangleq \sum_{s'' \in \sigma^{-1}(s')} T(s'' | s, a) \qquad \text{for all } s, s' \in \mathcal{S}, a \in \mathcal{A}. \qquad (7)$$

An exact MDP homomorphism provides a model equivalent abstraction [20]. Given an MDP homomorphism $h$, two state-action pairs $(s, a)$ and $(s', a')$ are called *$h$-equivalent* if $\sigma(s) = \sigma(s')$ and $\alpha_s(a) = \alpha_{s'}(a')$. Symmetries and MDP homomorphisms are connected in a natural way: If an MDP has symmetries $L_g$ and $K_g$, the above equations (4) and (5) hold. This means that we can define a corresponding MDP homomorphism, which we define next.

**Group-structured MDP Homomorphisms**   Specifically, for an MDP with symmetries, we can define an abstract state-action space, by mapping $(s, a)$ pairs to (a representative point of) their equivalence class $(\sigma(s), \alpha_s(a))$. That is, state-action pairs and their transformed version are mapped to the same abstract state in the reduced MDP:

$$(\sigma(s), \alpha_s(a)) = \big(\sigma(L_g[s]), \alpha_{L_g[s]}(K_g^s[a])\big) \quad \forall g \in G, s \in \mathcal{S}, a \in \mathcal{A} \qquad (8)$$

In this case, we call the resulting MDP homomorphism *group structured*. In other words, all the state-action pairs in an orbit defined by a group transformation are mapped to the same abstract state by a group-structured MDP homomorphism.

**Optimal Value Equivalence and Lifted Policies**   $h$-equivalent state-action pairs share the same optimal $Q$-value and optimal value function [30]. Furthermore, there exists an abstract optimal $Q$-value $\bar{Q}^*$ and abstract optimal value function $\bar{V}^*$, such that $Q^*(s, a) = \bar{Q}^*(\sigma(s), \alpha_s(a))$ and $V^*(s) = \bar{V}^*(\sigma(s))$. This is known as *optimal value equivalence* [30]. Policies can thus be optimized in the simpler abstract MDP. The optimal abstract policy $\bar{\pi}(\bar{a} | \sigma(s))$ can then be pulled back to the original MDP using a procedure called *lifting* [2]. The lifted policy is given in Equation 9. A lifted optimal abstract policy is also an optimal policy in the original MDP [30]. Note that while other lifted policies exist, we follow [30, 32] and choose the lifting that divides probability mass uniformly over the preimage:

$$\pi^\uparrow(a | s) \triangleq \frac{\bar{\pi}(\bar{a} | \sigma(s))}{|\{a \in \alpha_s^{-1}(\bar{a})\}|}, \qquad \text{for any } s \in \mathcal{S} \text{ and } a \in \alpha_s^{-1}(\bar{a}). \qquad (9)$$

## 3   Method

The focus of the next section is on the design of *MDP homomorphic networks*—policy networks and value networks obeying the MDP homomorphism. In the first section of the method, we show that any

policy network satisfying the MDP homomorphism property must be an equivariant neural network. In the second part of the method, we introduce a novel numerical technique for constructing group-equivariant networks, based on the transformation operators defining the equivalence state-action pairs under the MDP homomorphism.

## 3.1 Lifted Policies Are Invariant

Lifted policies in symmetric MDPs with group-structured symmetries are invariant under the group of symmetries. Consider the following: Take an MDP with symmetries defined by transformation operators $(L_g, K_g^s)$ for $g \in G$. Now, if we take $s' = L_g[s]$ and $a' = K_g^s[a]$ for any $g \in G$, $(s', a')$ and $(s, a)$ are h-equivalent under the corresponding MDP homomorphism $h = (\sigma, \{\alpha_s | s \in \mathcal{S}\})$. So

$$\pi^\uparrow(a|s) = \frac{\bar{\pi}(\alpha_s(a)|\sigma(s))}{|\{a \in \alpha_s^{-1}(\bar{a})\}|} = \frac{\bar{\pi}(\alpha_{s'}(a')|\sigma(s'))}{|\{a' \in \alpha_{s'}^{-1}(\bar{a})\}|} = \pi^\uparrow(a'|s'), \tag{10}$$

for all $s \in \mathcal{S}, a \in \mathcal{A}$ and $g \in G$. In the first equality we have used the definition of the lifted policy. In the second equality, we have used the definition of $h$-equivalent state-action pairs, where $\sigma(s) = \sigma(L_g(s))$ and $\alpha_s(a) = \alpha_{s'}(a')$. In the third equality, we have reused the definition of the lifted policy. Thus we see that, written in this way, the lifted policy is invariant under state-action transformations $(L_g, K_g^s)$. This equation is very general and applies for all group-structured state-action transformations. For a finite action space, this statement of invariance can be re-expressed as a statement of equivariance, by considering the vectorized policy.

**Invariant Policies On Finite Action Spaces Are Equivariant Vectorized Policies** For convenience we introduce a vector of probabilities for each of the discrete actions under the policy

$$\boldsymbol{\pi}(s) \triangleq [\pi(a_1|s), \quad \pi(a_2|s), \quad ..., \quad \pi(a_N|s)]^\top, \tag{11}$$

where $a_1, ..., a_N$ are the $N$ possible discrete actions in action space $\mathcal{A}$. The action transformation $K_g^s$ maps actions to actions invertibly. Thus applying an action transformation to the vectorized policy permutes the elements. We write the corresponding permutation matrix as $\mathbf{K}_g$. Note that

$$\mathbf{K}_g^{-1}\boldsymbol{\pi}(s) \triangleq [\pi(K_g^s[a_1]|s), \quad \pi(K_g^s[a_2]|s), \quad ..., \quad \pi(K_g^s[a_N]|s)]^\top, \tag{12}$$

where writing the inverse $\mathbf{K}_g^{-1}$ instead of $\mathbf{K}_g$ is required to maintain the property $\mathbf{K}_g\mathbf{K}_h = \mathbf{K}_{gh}$. The invariance of the lifted policy can then be written as $\boldsymbol{\pi}^\uparrow(s) = \mathbf{K}_g^{-1}\boldsymbol{\pi}^\uparrow(L_g[s])$, which can be rearranged to the equivariance equation

$$\mathbf{K}_g\boldsymbol{\pi}^\uparrow(s) = \boldsymbol{\pi}^\uparrow(L_g[s]) \qquad \text{for all } g \in G, s \in \mathcal{S}, a \in \mathcal{A}. \tag{13}$$

This equation shows that the lifted policy must satisfy an equivariance constraint. In deep learning, this has already been well-explored in the context of supervised learning [11, 12, 46, 47, 43]. Next, we present a novel way to construct such networks.

## 3.2 Building MDP Homomorphic Networks

Our goal is to build neural networks that follow Eq. 13; that is, we wish to find neural networks that are *equivariant* under a set of state and policy transformations. Equivariant networks are common in supervised learning [11, 12, 46, 47, 43, 41]. For instance, in semantic segmentation shifts and rotations of the input image result in shifts and rotations in the segmentation. A neural network consisting of only equivariant layers and non-linearities is equivariant as a whole, too[3] [11]. Thus, once we know how to build a single equivariant layer, we can simply stack such layers together. Note that this is true regardless of the representation of the group, i.e. this works for spatial transformations of the input, feature map permutations in intermediate layers, and policy transformations in the output layer. For the experiments presented in this paper, we use the same group representations for the intermediate layers as for the output, i.e. permutations. For finite groups, such as cyclic groups or permutations, pointwise nonlinearities preserve equivariance [11].

In the past, learnable equivariant layers were designed by hand for each transformation group individually [11, 12, 46, 47, 44, 43, 41]. This is time-consuming and laborious. Here we present a novel way to build learnable linear layers that satisfy equivariance automatically.

**Equivariant Layers** We begin with a single linear layer $\mathbf{z}' = \mathbf{W}\mathbf{z} + \mathbf{b}$, where $\mathbf{W} \in \mathbb{R}^{D_{\text{out}} \times D_{\text{in}}}$ and $\mathbf{b} \in \mathbb{R}^{D_{\text{in}}}$ is a bias. To simplify the math, we merge the bias into the weights so $\mathbf{W} \mapsto [\mathbf{W}, \mathbf{b}]$ and $\mathbf{z} \mapsto [\mathbf{z}, 1]^\top$. We denote the space of the augmented weights as $\mathcal{W}_{\text{total}}$. For a given pair of linear group transformation operators in matrix form $(\mathbf{L}_g, \mathbf{K}_g)$, where $\mathbf{L}_g$ is the input transformation and $\mathbf{K}_g$ is the output transformation, we then have to solve the equation

$$\mathbf{K}_g\mathbf{W}\mathbf{z} = \mathbf{W}\mathbf{L}_g\mathbf{z}, \qquad \text{for all } g \in G, \mathbf{z} \in \mathbb{R}^{D_{\text{in}}+1}. \tag{14}$$

Since this equation is true for all $\mathbf{z}$ we can in fact drop $\mathbf{z}$ entirely. Our task now is to find all weights $\mathbf{W}$ which satisfy Equation 14. We label this space of equivariant weights as $\mathcal{W}$, defined as

$$\mathcal{W} \triangleq \{\mathbf{W} \in \mathcal{W}_{\text{total}} \mid \mathbf{K}_g\mathbf{W} = \mathbf{W}\mathbf{L}_g, \text{ for all } g \in G\}, \tag{15}$$

again noting that we have dropped $\mathbf{z}$. To find the space $\mathcal{W}$ notice that for each $g \in G$ the constraint $\mathbf{K}_g\mathbf{W} = \mathbf{W}\mathbf{L}_g$ is in fact linear in $\mathbf{W}$. Thus, to find $\mathcal{W}$ we need to solve a set of linear equations in $\mathbf{W}$. For this we introduce a construction, which we call a *symmetrizer* $S(\mathbf{W})$. The symmetrizer is

$$S(\mathbf{W}) \triangleq \frac{1}{|G|} \sum_{g \in G} \mathbf{K}_g^{-1}\mathbf{W}\mathbf{L}_g. \tag{16}$$

$S$ has three important properties, of which proofs are provided in Appendix A. First, $S(\mathbf{W})$ is *symmetric* ($S(\mathbf{W}) \in \mathcal{W}$). Second, $S$ *fixes* any symmetric $\mathbf{W}$: ($\mathbf{W} \in \mathcal{W} \implies S(\mathbf{W}) = \mathbf{W}$). These properties show that $S$ projects arbitrary $\mathbf{W} \in \mathcal{W}_{\text{total}}$ to the equivariant subspace $\mathcal{W}$.

Since $\mathcal{W}$ is the solution set for a set of simultaneous linear equations, $\mathcal{W}$ is a linear subspace of the space of all possible weights $\mathcal{W}_{\text{total}}$. Thus each $\mathbf{W} \in \mathcal{W}$ can be parametrized as a linear combination of basis weights $\{\mathbf{V}_i\}_{i=1}^r$, where $r$ is the rank of the subspace and $\text{span}(\{\mathbf{V}_i\}_{i=1}^r) = \mathcal{W}$. To find as basis for $\mathbf{W}$, we take a Gram-Schmidt orthogonalization approach. We first sample weights in the total space $\mathcal{W}_{\text{total}}$ and then project them into the equivariant subspace with the symmetrizer. We do this for multiple weight matrices, which we then stack and feed through a singular value decomposition to find a basis for the equivariant space. This procedure is outlined in Algorithm 1. Any equivariant layer can then be written as a linear combination of bases

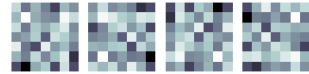

Figure 3: Example of 4-way rotationally symmetric filters.

$$\mathbf{W} = \sum_{i=1}^r c_i\mathbf{V}_i, \tag{17}$$

where the $c_i$'s are learnable scalar coefficients, $r$ is the rank of the equivariant space, and the matrices $\mathbf{V}_i$ are the basis vectors, formed from the reshaped right-singular vectors in the SVD. An example is shown in Figure 3. To run this procedure, all that is needed are the transformation operators $\mathbf{L}_g$ and $\mathbf{K}_g$. Note we do not need to know the explicit transformation matrices, but just to be able to perform the mappings $\mathbf{W} \mapsto \mathbf{W}\mathbf{L}_g$ and $\mathbf{W} \mapsto \mathbf{K}_g^{-1}\mathbf{W}$. For instance, some matrix $\mathbf{L}_g$ rotates an image patch, but we could equally implement $\mathbf{W}\mathbf{L}_g$ using a built-in rotation function. Code is available [4].

## 4 Experiments

We evaluated three flavors of MDP homomorphic network—an MLP, a CNN, and an equivariant feature extractor—on three RL tasks that exhibit group symmetry: CartPole, a grid world, and Pong.

**Algorithm 1** Equivariant layer construction

1: Sample $N$ weight matrices $\mathbf{W}_1, \mathbf{W}_2, ..., \mathbf{W}_N \sim \mathcal{N}(\mathbf{W}; \mathbf{0}, \mathbf{I})$ for $N \geq \dim(\mathcal{W}_{\text{total}})$
2: Symmetrize samples: $\bar{\mathbf{W}}_i = S(\mathbf{W}_i)$ for $i = 1, ..., N$
3: Vectorize samples and stack as $\bar{\mathbf{W}} = [\text{vec}(\bar{\mathbf{W}}_1), \text{vec}(\bar{\mathbf{W}}_2), ...]$
4: Apply SVD: $\bar{\mathbf{W}} = \mathbf{U}\boldsymbol{\Sigma}\mathbf{V}^\top$
5: Keep first $r = \text{rank}(\bar{\mathbf{W}})$ right-singular vectors (columns of $\mathbf{V}$) and unvectorize to shape of $\mathbf{W}_i$

---

Table 1: ENVIRONMENTS AND SYMMETRIES: We showcase a visual guide of the state and action spaces for each environment along with the effect of the transformations. Note, the symbols should not be taken to be hard mathematical statements, they are merely a visual guide for communication.

| Environment | | Space | Transformations |
|---|---|---|---|
| CartPole | $\mathcal{S}$ | $(x, \theta, \dot{x}, \dot{\theta})$ | $(x, \theta, \dot{x}, \dot{\theta}), (-x, -\theta, -\dot{x}, -\dot{\theta})$ |
| | $\mathcal{A}$ | $(\leftarrow, \rightarrow)$ | $(\leftarrow, \rightarrow), (\rightarrow, \leftarrow)$ |
| Grid World | $\mathcal{S}$ | $\{0, 1\}^{21 \times 21}$ | Identity, $\curvearrowright 90°$, $\curvearrowright 180°$, $\curvearrowright 270°$ |
| | $\mathcal{A}$ | $(\varnothing, \uparrow, \rightarrow, \downarrow, \leftarrow)$ | $(\varnothing, \uparrow, \rightarrow, \downarrow, \leftarrow), (\varnothing, \rightarrow, \downarrow, \leftarrow, \uparrow), (\varnothing, \downarrow, \leftarrow, \uparrow, \rightarrow), (\varnothing, \leftarrow, \uparrow, \rightarrow, \downarrow)$ |
| Pong | $\mathcal{S}$ | $\{0, ..., 255\}^{4 \times 80 \times 80}$ | Identity, reflect |
| | $\mathcal{A}$ | $(\varnothing, \varnothing, \uparrow, \downarrow, \uparrow, \downarrow)$ | $(\varnothing, \varnothing, \uparrow, \downarrow, \uparrow, \downarrow), (\varnothing, \varnothing, \downarrow, \uparrow, \downarrow, \uparrow)$ |

We use RLPYT [36] for the algorithms. Hyperparameters (and the range considered), architectures, and group implementation details are in the Supplementary Material. Code is available [5].

## 4.1 Environments

For each environment we show $\mathcal{S}$ and $\mathcal{A}$ with respective representations of the group transformations.

**CartPole** In the classic pole balancing task [3], we used a two-element group of reflections about the $y$-axis. We used OpenAI's Cartpole-v1 [7] implementation, which has a 4-dimensional observation vector: (cart position $x$, pole angle $\theta$, cart velocity $\dot{x}$, pole velocity $\dot{\theta}$). The (discrete) action space consists of applying a force left and right $(\leftarrow, \rightarrow)$. We chose this example for its simple symmetries.

**Grid world** We evaluated on a toroidal 7-by-7 predator-prey grid world with agent-centered coordinates. The prey and predator are randomly placed at the start of each episode, lasting a maximum of 100 time steps. The agent's goal is to catch the prey, which takes a step in a random compass direction with probability 0.15 and stands still otherwise. Upon catching the prey, the agent receives a reward of +1, and -0.1 otherwise. The observation is a $21 \times 21$ binary image identifying the position of the agent in the center and the prey in relative coordinates. See Figure 6a. This environment was chosen due to its four-fold rotational symmetry.

**Pong** We evaluated on the RLPYT [36] implementation of Pong. In our experiments, the observation consisted of the 4 last observed frames, with upper and lower margins cut off and downscaled to an $80 \times 80$ grayscale image. In this setting, there is a flip symmetry over the horizontal axis: if we flip the observations, the up and down actions also flip. A curious artifact of Pong is that it has duplicate (up, down) actions, which means that to simplify matters, we mask out the policy values for the second pair of (up, down) actions. We chose Pong because of its higher dimensional state space. Finally, for Pong we additionally compare to two data augmentation baselines: stochastic data augmentation, where for each state, action pair we randomly transform them or not before feeding them to the network, and the second an equivariant version of [16] and similar to [35], where both state and transformed state are input to the network. The output of the transformed state is appropriately transformed, and both policies are averaged.

## 4.2 Models

We implemented MDP homomorphic networks on top of two base architectures: MLP and CNN (exact architectures in Supplementary). We further experimented with an equivariant feature extractor, appended by a non-equivariant network, to isolate where equivariance made the greatest impact.

**Basis Networks** We call networks whose weights are linear combinations of basis weights *basis networks*. As an ablation study on all equivariant networks, we sought to measure the effects of the basis training dynamics. We compared an *equivariant* basis against a pure *nullspace* basis, i.e. an explicitly non-symmetric basis using the right-null vectors from the equivariant layer construction, and a *random* basis, where we skip the symmetrization step in the layer construction and use the full rank basis. Unless stated otherwise, we reduce the number of 'channels' in the basis networks compared to the regular networks by dividing by the square root of the group size, ending up with a comparable number of trainable parameters.

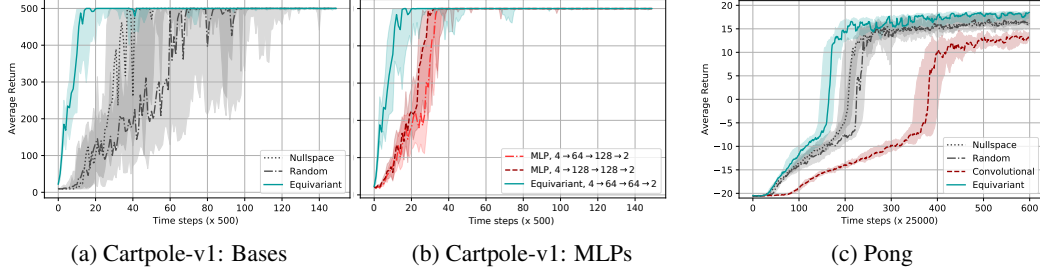

| (a) Cartpole-v1: Bases | (b) Cartpole-v1: MLPs | (c) Pong |
|---|---|---|

Figure 4: CARTPOLE: Trained with PPO, all networks fine-tuned over 7 learning rates. 25%, 50% and 75% quantiles over 25 random seeds shown. a) Equivariant, random, and nullspace bases. b) Equivariant basis, and two MLPs with different degrees of freedom. PONG: Trained with A2C, all networks tuned over 3 learning rates. 25%, 50% and 75% quantiles over 15 random seeds shown c) Equivariant, nullspace, and random bases, and regular CNN for Pong.

## 4.3   Results and Discussion

We show training curves for CartPole in 4a-4b, Pong in Figure 4c and for the grid world in Figure 6. Across all experiments we observed that the MDP homomorphic network outperforms both the non-equivariant basis networks and the standard architectures, in terms of convergence speed.

This confirms our motivations that building symmetry-preserving policy networks leads to faster convergence. Additionally, when compared to the data augmentation baselines in Figure 5, using equivariant networks is more beneficial. This is consistent with other results in the equivariance literature [4, 42, 44, 46]. While data augmentation can be used to create a larger dataset by exploiting symmetries, it does not directly lead to effective parameter sharing (as our approach does). Note, in Pong we only train the first 15 million frames to highlight the difference in the beginning; in constrast, a typical training duration is 50-200 million frames [25, 36].

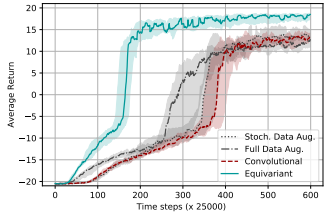

Figure 5: Data augmentation comparison on Pong.

For our ablation experiment, we wanted to control for the introduction of bases. It is not clear *a priori* that a network with a basis has the same gradient descent dynamics as an equivalent 'basisless' network. We compared equivariant, non-equivariant, and random bases, as mentioned above. We found the equivariant basis led to the fastest convergence. Figures 4a and 4c show that for CartPole and Pong the nullspace basis converged faster than the random basis. In the grid world there was no clear winner between the two. This is a curious result, requiring deeper investigation in a follow-up.

For a third experiment, we investigated what happens if we sacrifice complete equivariance of the policy. This is attractive because it removes the need to find a transformation operator for a flattened output feature map. Instead, we only maintained an equivariant feature extractor, compared against a basic CNN feature extractor. The networks built on top of these extractors were MLPs. The results, in Figure 4c, are two-fold: 1) Basis feature extractors converge faster than standard CNNs, and 2) the equivariant feature extractor has fastest convergence. We hypothesize the equivariant feature extractor is fastest as it is easiest to learn an equivariant policy from equivariant features.

We have additionally compared an equivariant feature extractor to a regular convolutional network on the Atari game Breakout, where the difference between the equivariant network and the regular network is much less pronounced. For details, see Appendix C.

## 5   Related Work

Past work on MDP homomorphisms has often aimed at discovering the map itself based on knowledge of the transition and reward function, and under the assumption of enumerable state spaces [30, 31, 32, 38]. Other work relies on learning the map from sampled experience from the MDP [39, 6, 23]. Exactly computing symmetries in MDPs is graph isomorphism complete [27] even with full knowledge of the MDP dynamics. Rather than assuming knowledge of the transition and reward function, and small and enumerable state spaces, in this work we take the inverse view: we assume that we have an easily identifiable transformation of the joint state–action space and exploit this knowledge

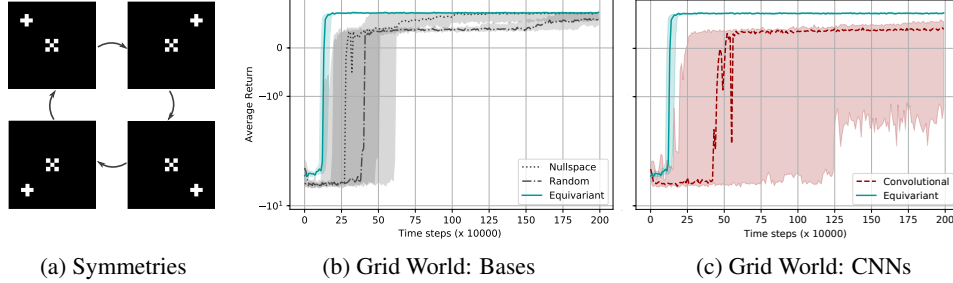

| (a) Symmetries | (b) Grid World: Bases | (c) Grid World: CNNs |

Figure 6: GRID WORLD: Trained with A2C, all networks fine-tuned over 6 learning rates. 25%, 50% and 75% quantiles over 20 random seeds shown. a) showcase of symmetries, b) Equivariant, nullspace, and random bases c) plain CNN and equivariant CNN.

to learn more efficiently. Exploiting symmetries in deep RL has been previously explored in the game of Go, in the form of symmetric filter weights [33, 8] or data augmentation [35]. Other work on data augmentation increases sample efficiency and generalization on well-known benchmarks by augmenting existing data points state transformations such as random translations, cutout, color jitter and random convolutions [16, 9, 17, 19]. In contrast, we encode symmetries into the neural network weights, leading to more parameter sharing. Additionally, such data augmentation approaches tend to take the *invariance* view, augmenting existing data with state transformations that leave the state's Q-values intact [16, 9, 17, 19] (the exception being [21] and [24], who augment trajectories rather than just states). Similarly, permutation invariant networks are commonly used in approaches to multi-agent RL [37, 22, 15]. We instead take the *equivariance* view, which accommodates a much larger class of symmetries that includes transformations on the action space. Abdolhosseini et al. [1] have previously manually constructed an equivariant network for a single group of symmetries in a single RL problem, namely reflections in a bipedal locomotion task. Our MDP homomorphic networks allow for automated construction of networks that are equivariant under arbitrary discrete groups and are therefore applicable to a wide variety of problems.

From an equivariance point-of-view, the automatic construction of equivariant layers is new. [12] comes close to specifying a procedure, outlining the system of equations to solve, but does not specify an algorithm. The basic theory of group equivariant networks was outlined in [11, 12] and [10], with notable implementations to 2D roto-translations on grids [46, 43, 41] and 3D roto-translations on grids [45, 44, 42]. All of these works have relied on hand-constructed equivariant layers.

# 6   Conclusion

This paper introduced MDP homomorphic networks, a family of deep architectures for reinforcement learning problems where symmetries have been identified. MDP homomorphic networks tie weights over symmetric state-action pairs. This weight-tying leads to fewer degrees-of-freedom and in our experiments we found that this translates into faster convergence. We used the established theory of MDP homomorphisms to motivate the use of equivariant networks, thus formalizing the connection between equivariant networks and symmetries in reinforcement learning. As an innovation, we also introduced the first method to automatically construct equivariant network layers, given a specification of the symmetries in question, thus removing a significant implementational obstacle. For future work, we want to further understand the symmetrizer and its effect on learning dynamics, as well as generalizing to problems that are not fully symmetric.

# 7   Acknowledgments and Funding Disclosure

Elise van der Pol was funded by Robert Bosch GmbH. Daniel Worrall was funded by Philips. F.A.O. received funding from the European Research Council (ERC) under the European Union's Horizon 2020 research and innovation programme (grant agreement No. 758824 —INFLUENCE). Max Welling reports part-time employment at Qualcomm AI Research.

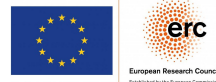

# 8 Broader Impact

The goal of this paper is to make (deep) reinforcement learning techniques
more efficient at solving Markov decision processes (MDPs) by making use of prior knowledge about symmetries. We do not expect the particular algorithm we develop to lead to immediate societal risks. However, Markov decision processes are very general, and can e.g. be used to model problems in autonomous driving, smart grids, and scheduling. Thus, solving such problems more efficiently can in the long run cause positive or negative societal impact.

For example, making transportation or power grids more efficient, thereby making better use of scarce resources, would be a significantly positive impact. Other potential applications, such as in autonomous weapons, pose a societal risk [28]. Like many AI technologies, when used in automation, our technology can have a positive impact (increased productivity) and a negative impact (decreased demand) on labor markets.

More immediately, control strategies learned using RL techniques are hard to verify and validate. Without proper precaution (e.g. [40]), employing such control strategies on physical systems thus run the risk of causing accidents involving people, e.g. due to reward misspecification, unsafe exploration, or distributional shift [2].

## Footnotes

[1]Specifically group equivariant networks are universal approximators to functions symmetric under linear representations of compact groups.

[2]Note that we use the terminology *lifting* to stay consistent with [30].

[3]See Appendix B for more details.

[4]https://github.com/ElisevanderPol/symmetrizer/

[5]https://github.com/ElisevanderPol/mdp-homomorphic-networks

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
