[Supplementary Material]

# Supplementary Material for
# MDP Homomorphic Networks:
# Group Symmetries in Reinforcement Learning

**Elise van der Pol**
UvA-Bosch Deltalab
University of Amsterdam
e.e.vanderpol@uva.nl

**Daniel E. Worrall**
Philips Lab
University of Amsterdam
d.e.worrall@uva.nl

**Herke van Hoof**
UvA-Bosch Deltalab
University of Amsterdam
h.c.vanhoof@uva.nl

**Frans A. Oliehoek**
Department of Intelligent Systems
Delft University of Technology
f.a.oliehoek@tudelft.nl

**Max Welling**
UvA-Bosch Deltalab
University of Amsterdam
m.welling@uva.nl

## A    The Symmetrizer

In this section we prove three properties of the symmetrizer: the symmetric property ($S(\mathbf{W}) \in \mathcal{W}$ for all $\mathbf{W} \in \mathcal{W}_{\text{total}}$ ), the fixing property ($\mathbf{W} \in \mathcal{W} \implies S(\mathbf{W}) = \mathbf{W}$) , and the idempotence property ($S(S(\mathbf{W})) = S(\mathbf{W})$ for all $\mathbf{W} \in \mathcal{W}_{\text{total}}$).

**The Symmetric Property**    Here we show that the symmetrizer $S$ maps matrices $\mathbf{W} \in \mathcal{W}_{\text{total}}$ to equivariant matrices $S(\mathbf{W}) \in \mathcal{W}$. For this, we show that a symmetrized weight matrix $S(\mathbf{W})$ from Equation 16 satisfies the equivariance constraint of Equation 14.

*The symmetric property.*  We begin by recalling the equivariance constraint

$$\mathbf{K}_g \mathbf{W} \mathbf{z} = \mathbf{W} \mathbf{L}_g \mathbf{z}, \qquad \text{for all } g \in G, \mathbf{z} \in \mathbb{R}^{D_{\text{in}}+1}. \tag{1}$$

Now note that we can drop the dependence on $\mathbf{z}$, since this equation is true for all $\mathbf{z}$. At the same time, we left-multiply both sides of this equation by $\mathbf{K}_g^{-1}$, which is possible because group representations are invertible. This results in the following set of equations

$$\mathbf{W} = \mathbf{K}_g^{-1} \mathbf{W} \mathbf{L}_g, \qquad \text{for all } g \in G. \tag{2}$$

Any $\mathbf{W}$ satisfying this equation satisfies Equation 1 and is thus a member of $\mathcal{W}$. To show that $S(\mathbf{W})$ is a member of $\mathcal{W}$, we thus would need show that $S(\mathbf{W}) = \mathbf{K}_g^{-1} S(\mathbf{W}) \mathbf{L}_g$ for all $\mathbf{W} \in \mathcal{W}_{\text{total}}$ and

$g \in G$. This can be shown as follows:

$$\mathbf{K}_g^{-1} S(\mathbf{W})\mathbf{L}_g = \mathbf{K}_g^{-1}\left(\frac{1}{|G|}\sum_{h \in G}\mathbf{K}_h^{-1}\mathbf{W}\mathbf{L}_h\right)\mathbf{L}_g \quad \text{substitute } S(\mathbf{W}) = \mathbf{K}_g^{-1} S(\mathbf{W})\mathbf{L}_g \tag{3}$$

$$= \frac{1}{|G|}\sum_{h \in G}\mathbf{K}_g^{-1}\mathbf{K}_h^{-1}\mathbf{W}\mathbf{L}_h\mathbf{L}_g \tag{4}$$

$$= \frac{1}{|G|}\sum_{h \in G}\mathbf{K}_{hg}^{-1}\mathbf{W}\mathbf{L}_{hg} \qquad \text{representation definition: } \mathbf{L}_h\mathbf{L}_h = \mathbf{L}_{hg} \tag{5}$$

$$= \frac{1}{|G|}\sum_{g'g^{-1} \in G}\mathbf{K}_{g'}^{-1}\mathbf{W}\mathbf{L}_{g'} \qquad \text{change of variables } g' = hg, h = g'g^{-1} \tag{6}$$

$$= \frac{1}{|G|}\sum_{g' \in Gg}\mathbf{K}_{g'}^{-1}\mathbf{W}\mathbf{L}_{g'} \qquad g'g^{-1} \in G \iff g' \in Gg \tag{7}$$

$$= \frac{1}{|G|}\sum_{g' \in G}\mathbf{K}_{g'}^{-1}\mathbf{W}\mathbf{L}_{g'} \qquad G = Gg \tag{8}$$

$$= S(\mathbf{W}) \qquad \text{definition of symmetrizer.} \tag{9}$$

Thus we see that $S(\mathbf{W})$ satisfies the equivariance constraint, which implies that $S(\mathbf{W}) \in \mathcal{W}$. ∎

**The Fixing Property** For the symmetrizer to be useful, we need to make sure that its range covers the equivariant subspace $\mathcal{W}$, and not just a subset of it; that is, we need to show that

$$\mathcal{W} = \{S(\mathbf{W}) \in \mathcal{W}|\mathbf{W} \in \mathcal{W}_{\text{total}}\}. \tag{10}$$

We show this by picking a matrix $\mathbf{W} \in \mathcal{W}$ and showing that $\mathbf{W} \in \mathcal{W} \implies S(\mathbf{W}) = \mathbf{W}$.

*The fixing property.* We begin by assuming that $\mathbf{W} \in \mathcal{W}$, then

$$S(\mathbf{W}) = \frac{1}{|G|}\sum_{g \in G}\mathbf{K}_g^{-1}\mathbf{W}\mathbf{L}_g \qquad \text{definition} \tag{11}$$

$$= \frac{1}{|G|}\sum_{g \in G}\mathbf{K}_g^{-1}\mathbf{K}_g\mathbf{W} \qquad \mathbf{W} \in \mathcal{W} \iff \mathbf{K}_g\mathbf{W} = \mathbf{W}\mathbf{L}_g, \forall g \in G \tag{12}$$

$$= \frac{1}{|G|}\sum_{g \in G}\mathbf{W} \tag{13}$$

$$= \mathbf{W} \tag{14}$$

This means that the symmetrizer leaves the equivariant subspace invariant. In fact, the statement we just showed is stronger in saying that each point in the equivariant subspace is unaltered by the symmetrizer. In the language of group theory we say that subspace $\mathcal{W}$ *is fixed under* $G$. Since $S : \mathcal{W}_{\text{total}} \to \mathcal{W}$ and there exist matrices $\mathbf{W}$ such that for every $\mathbf{W} \in \mathcal{W}$, $S(\mathbf{W}) = \mathbf{W}$, we have shown that

$$\mathcal{W} = \{S(\mathbf{W}) \in \mathcal{W}|\mathbf{W} \in \mathcal{W}_{\text{total}}\}. \tag{15}$$

∎

**The Idempotence Property** Here we show that the symmetrizer $S(\mathbf{W})$ from Equation 16 is idempotent, $S(S(\mathbf{W}))$.

*The idempotence property.* Recall the definition of the symmetrizer

$$S(\mathbf{W}) = \frac{1}{|G|}\sum_{g \in G}\mathbf{K}_g^{-1}\mathbf{W}\mathbf{L}_g. \tag{16}$$

Now let's expand $S(S(\mathbf{W}))$:

$$S(S(\mathbf{W})) = S\left(\frac{1}{|G|}\sum_{h\in G}\mathbf{K}_h^{-1}\mathbf{W}\mathbf{L}_h\right) \tag{17}$$

$$= \frac{1}{|G|}\sum_{g\in G}\mathbf{K}_g^{-1}\left(\frac{1}{|G|}\sum_{h\in G}\mathbf{K}_h^{-1}\mathbf{W}\mathbf{L}_h\right)\mathbf{L}_g \tag{18}$$

$$= \frac{1}{|G|}\sum_{g\in G}\left(\frac{1}{|G|}\sum_{h\in G}\mathbf{K}_g^{-1}\mathbf{K}_h^{-1}\mathbf{W}\mathbf{L}_h\mathbf{L}_g\right) \quad\text{linearity of sum} \tag{19}$$

$$= \frac{1}{|G|}\sum_{g\in G}\left(\frac{1}{|G|}\sum_{h\in G}\mathbf{K}_{hg}^{-1}\mathbf{W}\mathbf{L}_{hg}\right) \quad\text{definition of group representations} \tag{20}$$

$$= \frac{1}{|G|}\sum_{g\in G}\left(\frac{1}{|G|}\sum_{g'g^{-1}\in G}\mathbf{K}_{g'}^{-1}\mathbf{W}\mathbf{L}_{g'}\right) \quad\text{change of variables } g' = hg \tag{21}$$

$$= \frac{1}{|G|}\sum_{g\in G}\left(\frac{1}{|G|}\sum_{g'\in Gg}\mathbf{K}_{g'}^{-1}\mathbf{W}\mathbf{L}_{g'}\right) \quad g'g^{-1}\in G \iff g'\in Gg \tag{22}$$

$$= \frac{1}{|G|}\sum_{g\in G}\left(\frac{1}{|G|}\sum_{g'\in G}\mathbf{K}_{g'}^{-1}\mathbf{W}\mathbf{L}_{g'}\right) \quad Gg = G \tag{23}$$

$$= \frac{1}{|G|}\sum_{g'\in G}\mathbf{K}_{g'}^{-1}\mathbf{W}\mathbf{L}_{g'} \quad\text{sum over constant} \tag{24}$$

$$= S(\mathbf{W}) \tag{25}$$

Thus we see that $S(\mathbf{W})$ satisfies the equivariance constraint, which implies that $S(\mathbf{W}) \in \mathcal{W}$.  ∎

## B  Experimental Settings

### B.1  Designing representations

In the main text we presented a method to construct a space of intertwiners $\mathcal{W}$ using the symmetrizer. This relies on us already having chosen specific representations/transformation operators for the input, the output, and for every intermediate layer of the MDP homomorphic networks. While for the input space (state space) and output space (policy space), these transformation operators are easy to define, *it is an open question how to design a transformation operator for the intermediate layers* of our networks. Here we give some rules of thumb that we used, followed by the specific transformation operators we used in our experiments.

For each experiment we first identified the group $G$ of transformations. In every case, this was a finite group of size $|G|$, where the size is the number of elements in the group (number of distinct transformation operators). For example, a simple flip group as in Pong has two elements, so $|G| = 2$. Note that the group size $|G|$ does not necessarily equal the size of the transformation operators, whose size is determined by the dimensionality of the input/activation layer/policy.

**Stacking Equivariant Layers**  If we stack equivariant layers, the resulting network is equivariant as a whole too [1]. To see that this is the case, consider the following example. Assume we have network $f$, consisting of layers $f_1$ and $f_2$, which satisfy the layer-wise equivariance constraints:

$$P_g[f_1(x)] = f_1(L_g[x]) \tag{26}$$
$$K_g[f_2(x)] = f_2(P_g[x]) \tag{27}$$

With $K_g$ the output transformation of the network, $L_g$ the input transformation, and $P_g$ the intermediate transformation. Now,

$$K_g[f(x)] = K_g[f_2(f_1(x))] \tag{28}$$
$$= f_2(P_g[f_1(x)] \qquad (f_2 \text{ equivariance constraint}) \tag{29}$$
$$= f_2(f_1(L_g[x])) \qquad (f_1 \text{ equivariance constraint}) \tag{30}$$
$$= f(L_g[x]) \tag{31}$$

and so the whole network $f$ is equivariant with regards to the input transformation $L_g$ and the output transformation $K_g$. Note that this depends on the intermediate representation $P_g$ being shared between layers, i.e. $f_1$'s output transformation is the same as $f_2$'s input transformation.

**MLP-structured networks** For MLP-structured networks (CartPole), typically the activations have shape [`batch_size`, `num_channels`]. Instead we used a shape of [`batch_size`, `num_channels`, `representation_size`], where for the intermediate layers `representation_size=|G|+1` (we have a +1 because of the bias). The transformation operators we then apply to the activations is the set of permutations for group size $|G|$ appended with a 1 on the diagonal for the bias, acting on this last 'representation dimension'. Thus a forward pass of a layer is computed as

$$\mathbf{y}_{b,c_{\text{out}},r_{\text{out}}} = \sum_{c_{\text{in}}=1}^{\texttt{num\_channels}} \sum_{r_{\text{in}}=1}^{|G|+1} \mathbf{z}_{b,c_{\text{in}},r_{\text{in}}} \mathbf{W}_{c_{\text{out}},r_{\text{out}},c_{\text{in}},r_{\text{in}}} \tag{32}$$

where

$$\mathbf{W}_{c_{\text{out}},r_{\text{out}},c_{\text{in}},r_{\text{in}}} = \sum_{i=1}^{\text{rank}(\mathcal{W})} c_{i,c_{\text{out}},c_{\text{in}}} \mathbf{V}_{i,r_{\text{out}},r_{\text{in}}}. \tag{33}$$

**CNN-structured networks** For CNN-structured networks (Pong and Grid World), typically the activations have shape [`batch_size`, `num_channels`, `height`, `width`]. Instead we used a shape of [`batch_size`, `num_channels`, `representation_size`, `height`, `width`], where for the intermediate layers `representation_size=|G|+1`. The transformation operators we apply to the input of the layer is a spatial transformation on the `height`, `width` dimensions and a permutation on the `representation` dimension. This is because in the intermediate layers of the network the activations do not only transform in space, but also along the representation dimensions of the tensor. The transformation operators we apply to the output of the layer is just a permutation on the `representation` dimension. Thus a forward pass of a layer is computed as

$$\mathbf{y}_{b,c_{\text{out}},r_{\text{out}},h_{\text{out}},w_{\text{out}}} = \sum_{c_{\text{in}}=1}^{\texttt{num\_channels}} \sum_{r_{\text{in}}=1}^{|G|+1} \sum_{h_{\text{in}},w_{\text{in}}} \mathbf{z}_{b,c_{\text{in}},r_{\text{in}},h_{\text{out}}+h_{\text{in}},w_{\text{out}}+w_{\text{in}}} \mathbf{W}_{c_{\text{out}},r_{\text{out}},c_{\text{in}},r_{\text{in}},h_{\text{in}},w_{\text{in}}} \tag{34}$$

where

$$\mathbf{W}_{c_{\text{out}},r_{\text{out}},c_{\text{in}},r_{\text{in}},h_{\text{in}},w_{\text{in}}} = \sum_{i=1}^{\text{rank}(\mathcal{W})} c_{i,c_{\text{out}},c_{\text{in}}} \mathbf{V}_{i,r_{\text{out}},r_{\text{in}},h_{\text{in}},w_{\text{in}}}. \tag{35}$$

Table 1: Final learning rates used in CartPole-v1 experiments.

| Equivariant | Nullspace | Random | MLP |
|---|---|---|---|
| 0.01 | 0.005 | 0.001 | 0.001 |

## B.2   Cartpole-v1

**Group Representations**   For states:

$$\mathbf{L}_{g_e} = \begin{pmatrix} 1 & 0 & 0 & 0 \\ 0 & 1 & 0 & 0 \\ 0 & 0 & 1 & 0 \\ 0 & 0 & 0 & 1 \end{pmatrix}, \mathbf{L}_{g_1} = \begin{pmatrix} -1 & 0 & 0 & 0 \\ 0 & -1 & 0 & 0 \\ 0 & 0 & -1 & 0 \\ 0 & 0 & 0 & -1 \end{pmatrix}$$

For intermediate layers and policies:

$$\mathbf{K}^{\pi}_{g_e} = \begin{pmatrix} 1 & 0 \\ 0 & 1 \end{pmatrix}, \mathbf{K}^{\pi}_{g_1} = \begin{pmatrix} 0 & 1 \\ 1 & 0 \end{pmatrix}$$

For values we require an invariant rather than equivariant output. This invariance is implemented by defining the output representations to be $|G|$ identity matrices of the desired output dimensionality. For predicting state values we required a 1-dimensional output, and we thus used $|G|$ 1-dimensional identity matrices, i.e. for value output $V$:

$$\mathbf{K}^{V}_{g_e} = (1), \mathbf{K}^{V}_{g_1} = (1)$$

**Hyperparameters**   For both the basis networks and the MLP, we used Xavier initialization. We trained PPO using ADAM on 16 parallel environments and fine-tuned over the learning rates $\{0.01, 0.05, 0.001, 0.005, 0.0001, 0.0003, 0.0005\}$ by running 25 random seeds for each setting, and report the best curve. The final learning rates used are shown in Table 1. Other hyperparameters were defaults in RLPYT [2], except that we turn off learning rate decay.

**Architecture**
Basis networks:

Listing 1: Basis Networks Architecture for CartPole-v1

```
1  BasisLinear(repr_in=4, channels_in=1, repr_out=2, channels_out=64)
2  ReLU()
3  BasisLinear(repr_in=2, channels_in=64, repr_out=2, channels_out=64)
4  ReLU()
5  BasisLinear(repr_in=2, channels_in=64, repr_out=2, channels_out=1)
6  BasisLinear(repr_in=2, channels_in=64, repr_out=1, channels_out=1)
```

First MLP variant:

Listing 2: First MLP Architecture for CartPole-v1

```
1  Linear(channels_in=1, channels_out=64)
2  ReLU()
3  Linear(channels_in=64, channels_out=128)
4  ReLU()
5  Linear(channels_in=128,  channels_out=1)
6  Linear(channels_in=128, channels_out=1)
```

Second MLP variant:

Listing 3: Second MLP Architecture for CartPole-v1

```
1  Linear(channels_in=1, channels_out=128)
2  ReLU()
3  Linear(channels_in=128, channels_out=128)
4  ReLU()
5  Linear(channels_in=128,  channels_out=1)
6  Linear(channels_in=128, channels_out=1)
```

Table 2: Final learning rates used in grid world experiments.

| Equivariant | Nullspace | Random | CNN |
|---|---|---|---|
| 0.001 | 0.003 | 0.001 | 0.003 |

## B.3  GridWorld

**Group Representations**  For states we use `numpy.rot90`. The stack of weights is rolled.

For the intermediate representations:

$$\mathbf{L}_{g_e} = \begin{pmatrix} 1 & 0 & 0 & 0 \\ 0 & 1 & 0 & 0 \\ 0 & 0 & 1 & 0 \\ 0 & 0 & 0 & 1 \end{pmatrix}, \mathbf{L}_{g_1} = \begin{pmatrix} 0 & 0 & 0 & 1 \\ 1 & 0 & 0 & 0 \\ 0 & 1 & 0 & 0 \\ 0 & 0 & 1 & 0 \end{pmatrix}, \mathbf{L}_{g_2} = \begin{pmatrix} 0 & 0 & 1 & 0 \\ 0 & 0 & 0 & 1 \\ 1 & 0 & 0 & 0 \\ 0 & 1 & 0 & 0 \end{pmatrix}, \mathbf{L}_{g_3} = \begin{pmatrix} 0 & 1 & 0 & 0 \\ 0 & 0 & 1 & 0 \\ 0 & 0 & 0 & 1 \\ 1 & 0 & 0 & 0 \end{pmatrix}$$

For the policies:

$$\mathbf{K}^{\pi}_{g_e} = \begin{pmatrix} 1 & 0 & 0 & 0 & 0 \\ 0 & 1 & 0 & 0 & 0 \\ 0 & 0 & 1 & 0 & 0 \\ 0 & 0 & 0 & 1 & 0 \\ 0 & 0 & 0 & 0 & 1 \end{pmatrix}, \mathbf{K}^{\pi}_{g_1} = \begin{pmatrix} 1 & 0 & 0 & 0 & 0 \\ 0 & 0 & 0 & 0 & 1 \\ 0 & 1 & 0 & 0 & 0 \\ 0 & 0 & 1 & 0 & 0 \\ 0 & 0 & 0 & 1 & 0 \end{pmatrix}, \mathbf{K}^{\pi}_{g_2} = \begin{pmatrix} 1 & 0 & 0 & 0 & 0 \\ 0 & 0 & 0 & 1 & 0 \\ 0 & 0 & 0 & 0 & 1 \\ 0 & 1 & 0 & 0 & 0 \\ 0 & 0 & 1 & 0 & 0 \end{pmatrix}, \mathbf{K}^{\pi}_{g_3} = \begin{pmatrix} 1 & 0 & 0 & 0 & 0 \\ 0 & 0 & 1 & 0 & 0 \\ 0 & 0 & 0 & 1 & 0 \\ 0 & 0 & 0 & 0 & 1 \\ 0 & 1 & 0 & 0 & 0 \end{pmatrix}$$

For the values:

$$\mathbf{K}^{V}_{g_e} = (1), \mathbf{K}^{V}_{g_1} = (1), \mathbf{K}^{V}_{g_2} = (1), \mathbf{K}^{V}_{g_3} = (1)$$

**Hyperparameters**  For both the basis networks and the CNN, we used He initialization. We trained A2C using ADAM on 16 parallel environments and fine-tuned over the learning rates $\{0.00001, 0.00003, 0.0001, 0.0003, 0.001, 0.003\}$ on 20 random seeds for each setting, and reporting the best curve. The final learning rates used are shown in Table 2. Other hyperparameters were defaults in RLPYT [2].

**Architecture**
Basis networks:

Listing 4: Basis Networks Architecture for GridWorld

```
BasisConv2d(repr_in=1, channels_in=1, repr_out=4, channels_out=⌊16/√4⌋,
            filter_size=(7, 7), stride=2, padding=0)
ReLU()
BasisConv2d(repr_in=4, channels_in=⌊16/√4⌋, repr_out=4, channels_out=⌊32/√4⌋,
            filter_size=(5, 5), stride=1, padding=0)
ReLU()
GlobalMaxPool()
BasisLinear(repr_in=4, channels_in=⌊32/√4⌋, repr_out=4, channels_out=⌊512/√4⌋)
ReLU()
BasisLinear(repr_in=4, channels_in=⌊512/√4⌋, repr_out=5, channels_out=1)
BasisLinear(repr_in=4, channels_in=⌊512/√4⌋, repr_out=1, channels_out=1)
```

CNN:

Listing 5: CNN Architecture for GridWorld

```
Conv2d(channels_in=1, channels_out=16,
       filter_size=(7, 7), stride=2, padding=0)
ReLU()
Conv2d(channels_in=16,channels_out=32,
       filter_size=(5, 5), stride=1, padding=0)
ReLU()
GlobalMaxPool()
```

```
8   Linear(channels_in=32, channels_out=512)
9   ReLU()
10  Linear(channels_in=512, channels_out=5)
11  Linear(channels_in=512, channels_out=1)
```

## B.4  Pong

**Group Representations**   For the states we use `numpy`'s indexing to flip the input, i.e.
`w = w[..., ::-1, :]`, then the permutation on the `representation` dimension of the weights
is a `numpy.roll`, since the group is cyclic.

For the intermediate layers:

$$\mathbf{L}_{g_e} = \begin{pmatrix} 1 & 0 \\ 0 & 1 \end{pmatrix}, \mathbf{L}_{g_1} = \begin{pmatrix} 0 & 1 \\ 1 & 0 \end{pmatrix}$$

**Hyperparameters**   For both the basis networks and the CNN, we used He initialization. We
trained A2C using ADAM on 4 parallel environments and fine-tuned over the learning rates
$\{0.0001, 0.0002, 0.0003\}$ on 15 random seeds for each setting, and reporting the best curve. The
learning rates to fine-tune over were selected to be close to where the baseline performed well in
preliminary experiments. The final learning rates used are shown in Table 3. Other hyperparameters
were defaults in RLPYT [2].

**Architecture**
Basis Networks:

Listing 6: Basis Networks Architecture for Pong

```
1   BasisConv2d(repr_in=1, channels_in=4, repr_out=2, channels_out=⌊16/√2⌋,
2                filter_size=(8, 8), stride=4, padding=0)
3   ReLU()
4   BasisConv2d(repr_in=2, channels_in=⌊16/√2⌋, repr_out=2, channels_out=⌊32/√2⌋,
5                filter_size=(5, 5), stride=2, padding=0)
6   ReLU()
7   Linear(channels_in=2816, channels_out=⌊512/√2⌋)
8   ReLU()
9   Linear(channels_in=⌊512/√2⌋, channels_out=6)
10  Linear(channels_in=⌊512/√2⌋, channels_out=1)
```

CNN:

Listing 7: CNN Architecture for GridWorld

```
1   Conv2d(channels_in=4, channels_out=16, filter_size=(8, 8), stride=4, padding=0)
2   ReLU()
3   Conv2d(channels_in=16,channels_out=32, filter_size=(5, 5), stride=2, padding=0)
4   ReLU()
5   Linear(channels_in=2048, channels_out=512)
6   ReLU()
7   Linear(channels_in=512, channels_out=6)
8   Linear(channels_in=512, channels_out=1)
```

Table 4: Learning rates used in Breakout experiments.

| Equivariant | CNN |
| --- | --- |
| 0.0002 | 0.0002 |

Figure 1: BREAKOUT: Trained with A2C, all networks fine-tuned over 9 learning rates. 25%, 50% and 75% quantiles over 14 random seeds shown.

## C Breakout Experiments

We evaluated the effect of an equivariant basis extractor on Breakout, compared to a baseline convolutional network. The hyperparameter settings and architecture were largely the same as those of Pong, except for the input group representation, a longer training time, and that we considered a larger range of learning rates. To ensure symmetric states, we remove the two small decorative blocks in the bottom corners.

**Group Representations** For the states we use `numpy`'s indexing to flip the input, i.e.
`w = w[..., :, ::-1]` (note the different axis than in Pong), then the permutation on the `representation` dimension of the weights is a `numpy.roll`, since the group is cyclic.

For the intermediate layers:

$$\mathbf{L}_{g_e} = \begin{pmatrix} 1 & 0 \\ 0 & 1 \end{pmatrix}, \mathbf{L}_{g_1} = \begin{pmatrix} 0 & 1 \\ 1 & 0 \end{pmatrix}$$

**Hyperparameters** We used He initialization. We trained A2C using ADAM on 4 parallel environments and fine-tuned over the learning rates $\{0.001, 0.005, 0.0001, 0.0002, 0.0003, 0.0004, 0.0005, 0.00001, 0.00005\}$ on 15 random seeds for each setting, and reporting the best curve. The final learning rates used are shown in Table 4. Other hyperparameters were defaults in RLPYT [2].

**Results** Figure 1 shows the result of the equivariant feature extractor versus the convolutional baseline. While we again see an improvement over the standard convolutional approach, the difference is much less pronounced than in CartPole, Pong or the grid world. It is not straightforward why. One factor could be that the equivariant feature extractor is not end-to-end MDP homomorphic. It instead outputs a type of MDP homomorphic state representations and learns a regular policy on top. As a result, the unconstrained final layers may negate some of the advantages of the equivariant feature extractor. This may be more of an issue for Breakout than Pong, since Breakout is a more complex game.

(a) Cartpole-v1: Bases

(b) Cartpole-v1: CNNs

Figure 2: CARTPOLE: Trained with PPO, all networks fine-tuned over 7 learning rates. 25%, 50% and 75% quantiles over 25 random seeds shown. a) Equivariant, random, and nullspace bases. b) Equivariant basis, and two MLPs with different degrees of freedom.

## D   Cartpole-v1 Deeper Network Results

We show the effect of training a deeper network – 4 layers instead of 2 – for CartPole-v1 in Figure 2. The performance of the regular depth networks in Figure 4b and the deeper networks in Figure 2 is comparable, except that for the regular MLP, the variance is much higher when using deeper networks.

## E   Bellman Equations

$$V^\pi(s) = \sum_{a \in \mathcal{A}} \pi(s, a) \left[ R(s, a) + \gamma \sum_{s' \in \mathcal{S}} T(s, a, s') V^\pi(s') \right] \tag{36}$$

$$Q^\pi(s, a) = R(s, a) + \gamma \sum_{s' \in \mathcal{S}} T(s, a, s') V^\pi(s'). \tag{37}$$