[Reviews · NeurIPS 2020]

Review 1

Summary and Contributions: This paper introduces a method to account for symmetries in the state/observation and action spaces of MDPs when constructing deep NN policies for RL. It provides a fairly general approach for constructing trainable deep NNs which are equivariant under known symmetries, and shows in experiments that incorporating this prior knowledge can accelerate learning.

Strengths: Strengths of this work include the contribution of a novel and fairly general-purpose method for constructing deep NNs possessing equivariant properties, including convolution layers. This is an important area of study, as it has remained difficult to imbue deep RL agents with prior knowledge of such MDP structures. The experiments demonstrate faster learning by equivariant networks for the cases studied.

Weaknesses: The main weakness of the paper is the simplicity and specificity of the experiment environments/tasks. The method would be much more convincing if applied to an RL task with a more challenging representation-learning component. Examples could include a more advanced Atari game, DMControl from vision, or even better some 3-D visual environment. A more general discussion of symmetries in other common research environments, and what kind of symmetries can/cannot be represented, would be useful. More discussion about why the nullspace- and random-bases learning methods outperform the baseline convolutional approach in Pong would be useful, as this already provides most of the gain in performance.

Correctness: Yes, the theoretical developments are constructed correctly, and the experiments compare fairly against baseline algorithms.

Clarity: Yes, the claims are stated clearly and the derivations are clearly and straightforwardly organized.

Relation to Prior Work: The paper does relate to several previous works on symmetry / equivariance, and does distinguish its own contribution for the automatic construction of equivariant layers.

Reproducibility: Yes

Additional Feedback: Does the procedure to produce equivariant bases ever become computationally expensive? Are there any approximations / trade-offs one might consider?


Review 2

Summary and Contributions: The paper presents a method of constructing neural network architectures which hardcode symmetries of MDPs (for example flipping cart pole left to right and also interchanging the actions leads to an equivalent environment) with the goal of learning more efficiently if such symmetries are known in advance. A main contribution is an automated method for finding a basis of matrices which obey a particular symmetry. Such bases are used to construct network layers which respect the symmetries of the problem as a weighted combination of basis components weighted by trainable parameters. Experiments demonstrate that on 3 RL toy problems using such an architecture speeds learning compared to conventional neural networks and randomly chosen bases.

Strengths: The basic idea is reasonable and clearly explained. The proposed method for discovering bases of symmetrical weights is interesting and as far as I know novel. The experiments seem reasonably well thought out and illustrate the merit of the proposed approach.

Weaknesses: The main paper only presents the construction of an equivariant linear layer leaving the construction of full neural networks which are used in the experiments to the appendix. Even in the appendix I found the explanation to be somewhat limited so I feel this should be clarified and expanded. Experiments are limited to simple toy domains, leaving open the question of extending to more difficult problems.

Correctness: Experiments appear relatively well executed, including a description in the appendix of how the learning rate was tuned for each method. However tuning over only 3 learning rates in pong seems insufficient to assume you have found close to the best learning rate for each method and, in general given many of the optimal learning rates fall at the boundaries of the tested range it’s not clear whether expanding the range of learning rates tested would further improve performance. Perhaps including sensetivity curves for the learning rates would help with this. Also, the apparent instability of the CNN in Figure 5 (c) leads me to believe this baseline may not be well optimized.

Clarity: Most of the paper is clear and the underlying concepts are well explained, however I feel that a lot of information important for understanding the results is in the appendix and not in the main body of the paper. All that is described in the main body is how to construct a linear equivariant layer and it isn’t clear to me how this translates into constructing an equivariant neural network. It is claimed that citation [9] in the paper indicates that once we know how to build a single equivariant layer, we can simply stack such layers together. But this seems like a somewhat different use case than the one that applies here as I believe the transformations used in [9] are all spatial while in the present work the final transformation is in policy space. Moreover in the appendix it states "it is an open question how to design a transformation operator for the intermediate layers" however this isn't really made clear in the main body and the paragraph at line 176 even seems to convey the opposite impression. While I feel I have a general idea of how this must work, including a more explicit description would help to improve the clarity of the paper. At the moment even the description in the appendix is not entirely clear to me. For example, in section B.2 why are only Ls defined for states and Ks for intermediate layers and the policy? And why is the right thing to do to use the same Ks for policies and intermediate layers? At the very least I feel the explanation of how the ideas from the main paper are extended to a full neural network should be expanded and clarified in the appendix. Ideally at least a concise summary should also be included in the main body.

Relation to Prior Work: I am not very familiar with the prior work on MDP homomorphisms or equivariant neural networks in general. With that said the relevant background seemed to be covered well and the contribution was clearly stated.

Reproducibility: Yes

Additional Feedback: In the caption of figure 4 (b) I believe CNNs should be MLPs. In Figure captions it says that 25% 50% and 75% quantiles are shown but I only see one set of error bars. Are these captions incorrect? Line 142: Equation 9 should be Equation 8? Line 155: should this really be for all g given you are talking about a specific $s^\prime$ and $a^\prime$? Line 167: “The action transformation $K_g^s$ maps actions to actions invertibly”. Is invertibility an assumption here? I can’t immediately see why it should need to be so. For the grid world environment could you not have also added in reflection to the symmetry group? Line 245: what is the motivation for using a null space basis here? Is the idea to explicitly test with a basis of matrices that are not in the symmetry group? Line 260: I think figure 5c is incorrectly referenced here Line 261: “for CartPole and Pong the nullspace basis converged faster than the random basis. In the grid world there was no clear winner between the two.” I don’t see this in the plots, the error bars appear to overlap in all cases, am I missing something? Update ===== After the authors rebuttal and discussion with other reviewers I will maintain my score and recommend acceptance. Though I remain sympathetic to some arguments for rejection, I do feel this work could benefit the community by leading to interesting discussion. If I could see the authors revised discussion of how the approach applies to a full neural network I may consider raising my score but as is I can only hope this will be improved in the final version. I also share reviewer #1’s sentiment that more investigation into why null-space and random bases methods lead to such significant gains for pong would be very useful. I disagree somewhat with the point in the author’s feed back that “Data augmentation can benefit RL because it encourages symmetries by increasing the dataset, on the other hand, equivariance enforces them, so the network does not need to learn the symmetries”. This is perhaps true for “Data aug. 1” baseline. But if I understand “Data aug. 2” correctly, it is in some sense enforcing the symmetry because only the averaged result is ever used. This leads me to believe that this explanation is not sufficient to explain the benefit of the proposed approach. In fact, this leads me to one other point, is the use bases really necessary for symmetrization? It seems to me you could just replace W with S(W) in the network itself to produce a symetrized weight matrix. If I’m not mistaken, the proposed technique of generating a basis results in having to store n*(n-d) parameters, where d is the number of degrees of freedom removed by enforcing the constraints (one basis matrix for each parameter in the original matrix minus the removed degrees of freedom). Simply using S(W) in the network would only require summing |G| copies of W, instead of (n-d). It would be would be interesting to see a discussion/comparison of these two approaches in terms of computational complexity of training since the overhead of operating on a basis for each weight matrix seem like it would be rather significant.


Review 3

Summary and Contributions: This paper investigates MDP homomorphism but instead of assuming knowledge of the transition and reward function, the paper assumes the existence of an easily identifiable transformation of the state-action space (e.g., symmetries in an image). The paper then introduces the concept of group equivariant networks, networks that are equivariant under a set of state and policy transformations. Finally, the paper also introduces a numerical algorithm for the automated construction of equivariant layers and it demonstrates its applicability in two small domains (cartpole and a gridworld) and one pixel-based environment (the Atari 2600 game Pong).

Strengths: It formalizes what is now becoming a standard practice in the deep RL community: data augmentation. It is always useful to see a principled presentation of a concept that is becoming so pervasive in the field. Importantly, this paper is really well-written despite covering non-trivial mathematical concepts. It is also great that the paper goes beyond the formalization and actually proposes a new type of network that explicitly captures these homomorphisms. An important point is that the paper also shows the scalability of the proposed approach in a pixel-based domain.

Weaknesses: My main criticism about this paper is with respect to its empirical evaluation. I don't think the paper provides enough evidence that equivariant networks are better than maybe simpler options to capture invariances/symmetries. Specifically, if I'm knowledgeable of the transformations that lead to invariance (or equivariance), should I use equivariant networks instead of data augmentation? I'd be curious to see how a "regular" network, fed with the different transformations of the input, would perform when compared to equivariant networks. Is this an experiment that was run and I missed it? It seems to me the paper focuses too much on the constraints induced by the networks but not so much on how to leverage such an information with regular networks. Simply feeding a standard agent more frames, from transformations, would be a meaningful baseline. Others that come to mind include cutout [Cobbe et al., 2019], random convolutions [Lee et al., 2020], random shifts [Kostrikov et al., 2020], random crop, and color jitter [Laskin et al., 2020]. Such an experiment would be authoritative evidence that equivariant networks are a good approach to deal with problems in which we know its symmetries. If it is on par with these methods it might not be that interesting since it is definitely more complex. I don't expect the authors to compare the proposed solution to all methods listed above, some of them are only on arXiv and are quite recent, but I thought a more comprehensive list would be more useful. Complete references are below. Finally, the new network requires some matrix inversions. This is a particularly expensive operation. Obviously, the paper has results on an Atari 2600 game, showing that the proposed approach scales up to this setting. It would be interesting to see a longer discussion about the scalability of the proposed idea though. References: I. Kostrikov, D. Yarats, and R. Fergus, “Image augmentation is all you need: Regularizing deep reinforcement learning from pixels,” CoRR, vol. abs/2004.13649, 2020. K. Cobbe, O. Klimov, C. Hesse, T. Kim, and J. Schulman, “Quantifying generalization in reinforcement learning,” in Proceedings of the International Conference on Machine Learning (ICML), 2019. M. Laskin, K. Lee, A. Stooke, L. Pinto, P. Abbeel, and A. Srinivas, “Reinforcement learning with augmented data,” CoRR, vol. abs/2004.14990, 2020. K. Lee, K. Lee, J. Shin, and H. Lee, “Network randomization: A simple technique for generalization in deep reinforcement learning,” in The International Conference on Learning Representations (ICLR), 2020.

Correctness: To the best of my knowledge, yes, this paper is correct.

Clarity: The paper is very well-written. If I had to make a suggestion, maybe it would be to be more clear about what the baselines used in the empirical evaluation correspond to. What are the current methods that correspond to each baseline? What would, for example, the standard DQN (or PPO) correspond to?

Relation to Prior Work: The paper thoroughly discusses prior work related to homomorphisms. As I mentioned earlier, I wish there was a longer discussion about data augmentation and the interplay between simply feeding a "regular" network augmented data / designing architectures that exploit invariances / equivariances. Aside from that, it would also be interesting to see a discussion about whether such an approach can improve generalization in RL and how it is related to bisimulation metrics.

Reproducibility: Yes

Additional Feedback: I'm recommending the acceptance of this paper because of the mathematical formalization it introduces and also because of the new principled architecture that naturally captures invariances. It is quite interesting and well-written. I'm not giving it a higher score because of its empirical evaluation. Unless I missed something, the empirical results demonstrate the scalability and efficacy of the proposed architecture but they don't provide us with an intuition of what to favour. Data augmentation might still be a simpler approach to the same problem. ----- Comments after the rebuttal ----- I want to thank the authors for their response. The results comparing the proposed method to data augmentation are quite interesting. After the discussion with the other reviewers, and the authors' response, I decided to keep my score as is. I think this is a good paper and it should be accepted at NeurIPS. Some of the concerns that prevented me from raising my score was a more thorough empirical evaluation, maybe in settings in which the symmetry is not as obvious as in the domains it was evaluated. What would happen if the method was applied to a setting where the constraints do not hold? How would this approach degrade? It would be interesting to see a discussion about this in the final version of the paper.


Review 4

Summary and Contributions: This paper presents a family of deep nets that can incorporate equivariance properties for deep reinforcement learning. The contributions are two-folded, (1) they formalized the relation between equivariance properties for deep RL using MDP homomorphism, and (2) they proposed a novel algorithm to build equivariant layers. Experiments were conducted on CartPole, grid world and Pong environments demonstrating that incorporating equivariance properties lead to faster convergence.

Strengths: 1. Overall, I find the paper easy to read and motivational. I agree with the authors that deep RL rarely include equivariance properties into their modeling and their work could be useful at improving the data efficiency of deep RL. 2. The notation and approach are described clearly. In particular, the parts charactering the relationship between MDP Homomorphisms and equivariance properties is interesting and novel. While the result is somewhat expected, I think it deserves contribution and benefits the community to express the relations this clear manner. 3. Aside from the contribution to RL, this work also proposes an approach to automatically build equivariant layers rather than handcrafting them, which has the potential for making equivariant layers more practical and possibly adapt more into RL systems. 4. Code is included with submission and reported the range of hyperparameters used.

Weaknesses: Here are some concerns with the paper: a. The presented empirical evidence could be strengthened. In particular, the environments of CartPole, gird world, and Pong, are relatively toy examples for RL. More challenging environments, e.g. other atari games, would have made the results more convincing. b. From my understanding, the baselines are without data augmentations. Would a data augmentation approach be just as effective? For example, a flip symmetry would only double the amount of data being processed. Data augmentation also does not require hand-constructed layers and is easy to implement. A comparison will further demonstrate the effectiveness of the proposed approach. c. Will the proposed approach run into scalability issues if G is large? For example, when G is a permutation group. Minor: d. Prior works have also explored equivariance properties in RL, e.g., [A, B], which use graph neural networks for permutation equivariance/invariant garunatess in multi-agent settings. e. Is there a typo at Line 118 for Q*? It should be Q*(s,a), e.g. line 139. [A] Liu, Iou-Jen, Raymond A. Yeh, and Alexander G. Schwing. "Pic: Permutation invariant critic for multi-agent deep reinforcement learning." Conference on Robot Learning. 2019 [B] Jiang, Jiechuan, et al. "Graph Convolutional Reinforcement Learning." International Conference on Learning Representations. 2020

Correctness: Briefly check the proofs, looks correct. No issues with their empirical methodology.

Clarity: Yes, the paper is very well written.

Relation to Prior Work: The related works section adequately discussed and contrasted their approach with prior works.

Reproducibility: Yes

Additional Feedback: Misc: Fig. 5 at the end of paper seems odd, might consider moving it from the bottom of the page. (After author feedback) Thanks for the detailed feedback. - The experiment on data augmentation addressed part of my concern. - Additional results with other environments would have raised my overall score. - It might be worth discussing in the paper that the approach may be expensive if the group size is large.

[Author Response · NeurIPS 2020]

We thank the reviewers for their feedback and suggestions. Below we will clarify the points raised by the reviewers.

**Novelty**. Our approach connects MDP homomorphisms with equivariant networks, introduces a novel way for constructing such networks and uses them to speed up learning in RL. We are encouraged by the positive feedback of the reviewers regarding the novelty and method.

**Scalability**. To **R1**, **R3** and **R4**: Scalability is not directly an issue, largely because the most expensive step, the equivariant basis construction, is performed only once, prior to training. To **R3**, regarding matrix inversions: The transformation matrices we used are orthogonal, so that we can take the cheap transpose. To **R1**: A truncated SVD could provide a reasonable approximation if the one-time cost of construction is prohibitive. To **R1** and **R4**: We do not encounter issues within the transformation groups we consider. For large groups such as permutation groups we note that the number of filters scales linearly with the size of $G$, as does the number of input channels for the filters. For very large weight matrices, finding the SVD is computationally expensive.

**Data augmentation**.     We thank **R3** and **R4** for suggesting additional comparisons to data augmentation. Per **R3** and **R4**'s suggestion, we ran two data augmentation baselines. The first data augmentation is designed to be a direct port of supervised learning to RL, akin to **R3**'s suggestion: Each state image is randomly transformed or not. If it is transformed, the output is correspondingly transformed. The second data augmentation is an equivariant version of (Kostrikov 2020), where both state and transformed state are input to the network. The output of the transformed state is appropriately transformed, and both policies are averaged. We show results on 4 random seeds for Pong in Figure 1. While data augmentation is beneficial in RL, our approach outperforms both variants. This is consistent with other results in the equivariance literature (see e.g. Worrall 2017, Winkels 2018, Bekkers 2018, Weiler 2018). Data augmentation can benefit RL because it *encourages* symmetries by increasing the dataset, on the other hand, equivariance *enforces* them, so the network does not need to learn the symmetries. We will incorporate the comparison and a discussion in the paper.

Figure 1: Data augmentation baselines for Pong.

**Network construction**. To **R2**, regarding ambiguity about network construction. We will improve the explanation in the appendix and add a short summary to the main paper, and examples will be included in the released code. To clarify, the representation of the group in the intermediate layers can be chosen arbitrarily. Our proposed solution works for any discrete group (as shown to work best in Weiler 2019), but other choices are definitely possible.

**Other environments**. To **R1**, **R2** and **R4**: We focus on CartPole, grid world and Pong, because these environments provide varying levels of complexity while still being compact enough to allow us to run a grid search and multiple baselines across environments with different observation spaces and symmetry groups. Our approach is in theory applicable to any RL problem that exhibits discrete group symmetry. Thus, this method is certainly applicable to more complex Atari games that exhibit symmetry. Based on the suggestion by **R1**, **R2**, **R4**, we are currently evaluating on Breakout, a more challenging Atari game. Experiments are currently running but exceed the length of the rebuttal period. To **R1**, our method is indeed also applicable to DM control for vision, as it exhibits flip symmetry.

**Clarifications**. To **R2**: We use nullspace/random baselines to show that equivariance is key to improving performance. To **R1**: While nullspace/random perform similar to the regular baseline for the other two environments they perform better on Pong. We expect that this may be related to different gradient dynamics when using basis networks, which could influence learning. In all cases, equivariance performs best. To **R2**: The range we considered for Pong was chosen as the baseline performed much worse at other learning rate ranges. We therefore searched in only this range to optimize our own method. We use 6 learning rates in a larger range for grid world in Figure 5c. To **R3**: We think our approach can be useful for generalization, for example by learning in a state and directly generalizing to its transformed versions. To **R2**: The action transformation is a group representation, it therefore must have invertibility.

We thank all reviewers for their time and efforts. We will incorporate the experiments and discussions, as well as typos, references and minor issues in the paper, and release all code.

[Meta-Review · NeurIPS 2020]

The paper proposes an approach for incorporating knowledge about symmetries or equivariances into neural network policies by providing a general purpose method for constructing network layers based on knowledge of the relevant transformations. The reviews are generally positive: Identifying effective ways of incorporating prior knowledge of this type into neural networks is an important research challenge that is of interest to the community. The proposed approach for constructing network layers seems novel, although there is some prior work that explores ways of exploiting such knowledge in particular application domains, or via alternative means such as data augmentation. (Much of this is cited in the paper.) An important caveat of the submission, remarked upon by all reviewers is the experimental evaluation. It is currently limited to simple scenarios with perfect symmetries which provide limited evidence of the utility of the approach in more complex / less idealized scenarios. Furthermore, important baselines such as data augmentation approaches were added only in the rebuttal. Although the paper is generally clearly written the explanation of how invariant multi-layer networks are constructed could be approved. The reviewers discussed the paper extensively. All reviewers think that the idea will be useful to the community and has the potential to spark a discussion. Disagreement remained, however, whether the experimental evaluation is sufficient in the current form. On balance all reviewers feel positive about the paper. The meta reviewer would strongly encourage the authors to incorporate the data augmentation baselines as well as the results for Breakout (and / or some other non-idealized domain) into the final version. The authors may want to include the following paper in their related work section: https://arxiv.org/pdf/1909.10707.pdf